# Do Vision Language Models infer human intention without visual perspective-taking? Towards a scalable "One-Image-Probe-All" dataset

## Abstract

At the core of understanding the knowledge grounding of Multimodal Large Language Models (MLLMs) are two key challenges: (1) ensuring fair comparability across concepts and (2) scaling multimodal datasets to reflect real-world complexity. This paper presents a solution through the **Omni-Perspective** benchmark, which scales the construction of a 5-level question-context-answers (QCAs) **from 1 real-world image**. This benchmark pertains to 3 concepts along the Theory-of-Mind (ToM) ability hierarchy in humans and is further divided into 10 fine-grained subdifficulties. Through inference tasks, complexity, and ablation analysis, we evaluate over 2,200 consolidated QCAs on 61 MLLMs. Our findings reveal a key observation: MLLMs mostly follow the human ToM grounding pathway with exception of level-2 perspective taking. Furthermore, this dataset enables nuanced analysis of how such observations change across varying difficulty levels, modalities, distractor logic, and prompt types.

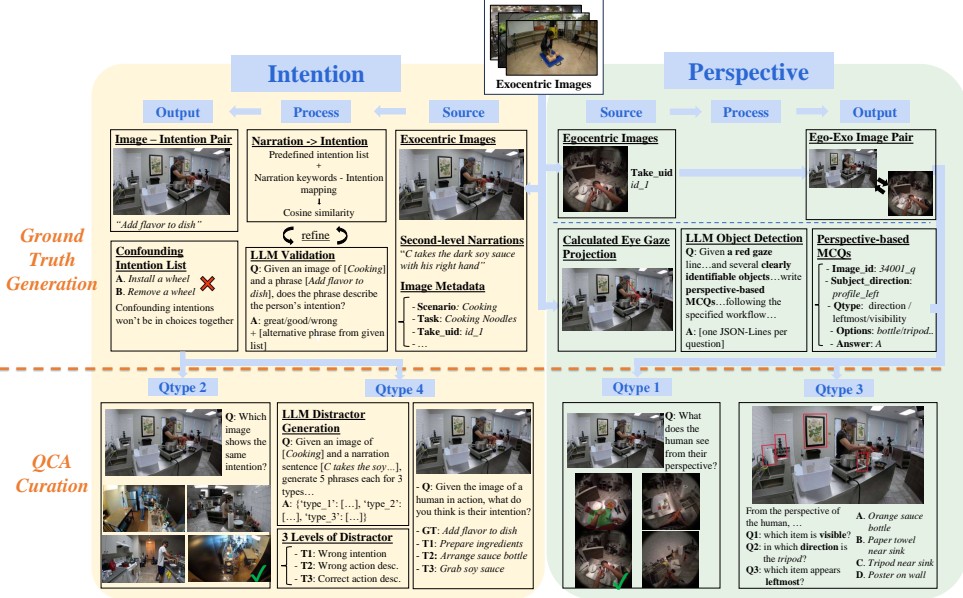

Figure 1: The scalable curation of Omni-perspective dataset

# 1 Introduction

Recent advances in Multimodal Large Language Models (MLLMs) have sparked growing interest in evaluating their capacity for complex reasoning grounded in both visual and linguistic inputs. However, rigorous assessment remains challenging due to the absence of scalable, cognitively structured benchmarks that support controlled, hierarchical, and comparative probing across diverse conceptual domains (Li et al., 2025). In this work, we address this gap by introducing a multi-image, hierarchical, and concept-controlled Question-Context-Answer (QCA) generation framework, designed to facilitate systematic evaluation of reasoning abilities across aligned tasks and cognitive levels. This framework enables the use of reusable image-intention pairs, supports fine-grained control over task difficulty, and allows for modular expansion to large-scale multimodal datasets—offering a generalizable solution for cognitively diagnostic evaluation.

A key application of this framework is the assessment of visual perspective-taking (VPT) in relation to Theory of Mind (ToM) capabilities (Premack and Woodruff, 1978; Barnes-Holmes et al., 2004; Schaafsma et al., 2015). VPT involves understanding what others see (Level 1, or VPT-1) and how they see it (Level 2, or VPT-2). Understood to be grounded in perspective-taking abilities, ToM entails modeling others' beliefs, goals, and intentions. These cognitive capacities develop in humans along a staged trajectory (Barnes-Holmes et al., 2004; Barsalou, 2008; Schurz et al., 2021), offering a natural scaffold for probing whether—and how—MLLMs internalize comparable representational structures (Sucholutsky et al., 2023).

While several benchmarks have explored vision-language reasoning, many are limited in either scope or ecological validity. For example, synthetic datasets such as CLEVR, CATER, and related benchmarks have demonstrated the utility of 3D scene modeling and controlled object manipulation for investigating compositional reasoning (Johnson et al., 2017; Girdhar and Ramanan, 2020). However, these datasets operate in highly idealized environments, characterized by clean object boundaries, minimal perceptual noise, and fully specified symbolic constraints. As a result, they tend to overestimate generalization: models trained and evaluated in these "lab-grade" settings often fail to transfer their reasoning capabilities to real-world scenes, where visual ambiguity, occlusion, temporal dynamics, and social intent are critical (Mitchell and Krakauer, 2023).

Benchmarks such as ALPRO and VQA-X expand the modality coverage and include real images or videos, but they often lack hierarchical cognitive task design or do not isolate the compositional demands of ToM-related inference. Moreover, overreliance on language priors can inflate performance in multimodal benchmarks even when visual inputs are ignored, undermining interpretability (Dongxu Li, 2022; Park et al., 2018).

To address these limitations, we propose Omni-Perspective, a cognitively motivated benchmark instantiated from our QCA generation framework. Built upon the rich, multimodal Ego-Exo4D dataset, Omni-Perspective includes over 2,200 curated QCAs structured around a six-level hierarchy that spans low-level spatial awareness to high-level belief reasoning. Each question is grounded in a shared image-intention pair and linked to a cognitive hypothesis, enabling both depth and comparability across reasoning types. Our scalable pipeline combines narration-intention mappings with GPT-4o-assisted refinement, allowing for high-quality annotation at scale without extensive manual labeling.

We evaluate 50+ MLLMs of varying modalities, sizes, and pretraining objectives, finding that while many models perform well on spatial reasoning, they falter on belief-based or intention-predictive tasks. This suggests a deviation from the developmental trajectory observed in human ToM, and motivates architectural or training-level interventions to improve grounding and inference capabilities.

In summary, this work makes three key contributions:

1. **A multi-modal probing framework** for scalable, hierarchical, and controlled Question-Context-Answer (QCA) generation, aligned with cognitive theory for systematic evaluation of multimodal reasoning.

2. **A controlled and hierarchical benchmark**, *Omni-Perspective*, designed to probe Theory of Mind (ToM) and visual perspective-taking abilities using real-world, multimodal visual data from naturalistic scenarios.

3. **An empirical analysis** revealing consistent ToM-related failure modes in state-of-the-art MLLMs, offering diagnostic insights and guiding principles for future model and training improvements.

## 2 Related Works

### 2.1 MLLM related

#### 2.1.1 Benchark

The field of Multi-modal Large Language Models (MLLMs) requires a comprehensive evaluation of their remarkable capabilities to ensure that their development is progressing on a correct and appropriate trajectory. Early benchmarks primarily focused on single tasks, such as VQA (Antol et al., 2015), OK-VQA (Marino et al., 2019), MSCOCO (Lin et al., 2015), OCR (Liu et al., 2023), and GQA (Hudson and Manning, 2019), but have become insufficient for thoroughly assessing the broad multimodal perception and reasoning abilities of LMMs. In response, more holistic evaluations have emerged, such as LAMM (Yin et al., 2024), MM-Vet (Yu et al., 2023), SEED-Bench (Li et al., 2024), and MMBench (Liu et al., 2024c), which cover a wider range of capabilities.

#### 2.1.2 Multi-modal Large Language Models

Recent advancements in multimodal learning have been largely driven by the unified modeling of visual and textual data using transformers (Li et al., 2019; Xu et al., 2023; Tan and Bansal, 2019; Alayrac et al., 2022; Radford et al., 2021). With the emergence of Large Language Models (LLMs), state-of-the-art (SOTA) Multi-modal Large Language Models (MLLMs) (Liu et al., 2024a; Li et al., 2023a) now integrate open-source LLMs (Touvron et al., 2023; Peng et al., 2023; Jiang et al., 2023), aligning visual features with the embedding space of LLMs (Li et al., 2023b).

To enhance open-ended conversational abilities, LLaVA (Liu et al., 2024a) introduces a method to distill the conversational capabilities of ChatGPT into MLLMs, resulting in a substantial performance boost. This approach has since become a standard procedure in the field (Wang et al., 2023; Bai et al., 2023; Gemini, 2023; Team, 2024; Sun et al., 2023; Li et al., 2022). As a result, MLLMs have demonstrated competitive performance in complex tasks requiring high-level perception and reasoning (Li et al., 2024; Liu et al., 2024a; Gemini, 2023; Fu et al., 2023; OpenAI, 2023), including spatial reasoning (Chen et al., 2024; Cai et al., 2024), character recognition (Mori et al., 1999), scene understanding (Cordts et al., 2016; Chen et al., 2017), action recognition (Jhuang et al., 2013; Herath et al., 2017), and prediction (Lan et al., 2014; Kong and Fu, 2022), often reaching near-human performance.

### 2.2 Visual perspective taking, Intentionality and Theory-of-Mind

The capacity to adopt another individual's visual perspective is widely recognized as a foundational component of social cognition and is considered a developmental precursor to theory of mind (ToM)—the ability to attribute mental states such as beliefs, intentions, and knowledge to oneself and others (Premack and Woodruff, 1978). While early research emphasized intention inference as central to ToM, more recent accounts have identified visual perspective taking (VPT) as a perceptual substrate supporting the emergence of mental state attribution. VPT is typically differentiated into two levels: Level-1 perspective taking (VPT-1) involves representing *what* another agent can see (i.e., which objects fall within their line of sight) whereas Level-2 perspective taking (VPT-2) entails representing *how* those objects appear from another spatial viewpoint, including their orientation and relative configuration (Kessler and Rutherford, 2010). Because VPT-2 requires mental transformations of one's egocentric reference frame—often instantiated through embodied simulation or motor imagery—it has been proposed as a particularly robust route to social understanding, even though such simulation is not strictly necessary for theory of mind reasoning in general (Hamilton et al., 2009; Gallese and Goldman, 1998; Barlassina and Gordon, 2017).

Beyond these two levels, several developmental models posit a graded trajectory in which perceptual perspective taking scaffolds increasingly abstract forms of social cognition. For example, Barnes-Holmes and colleagues propose a sequence extending from recognition of differing viewpoints to inferential use of perceptual access for epistemic judgments, prediction of actions based on true

beliefs, and ultimately the attribution of behavior based on false beliefs (Barnes-Holmes et al., 2004). Although terminological distinctions vary across frameworks, similar hierarchical structures were long proposed in traditional Piagetian theories of cognitive development (Piaget and Inhelder, 1969) and have since been elaborated in contemporary neurocognitive models that integrate perspective taking, empathy, and mental state attribution along continuous processing gradients (Schurz et al., 2021). Converging evidence from theoretical analyses suggests that tasks classified as measuring theory of mind in fact engage a distributed set of perceptual, inferential, and executive systems as opposed to being targeting a monolithic construct (Schaafsma et al., 2015; Quesque and Rossetti, 2020; Barresi and Moore, 1996). These perspectives collectively support the view that higher-order social reasoning emerges through the gradual abstraction of perceptual and embodied capacities like visual perspective taking.

This developmental progression aligns with the theoretical framework of grounded cognition, which posits that high-level cognitive functions are constitutively supported by sensorimotor systems evolved for real-world interaction (Barsalou, 2008; Gallese, 2007). Accordingly, visual perspective taking offers a principled pathway through which embodied simulation mechanisms give rise to abstract representations of others' mental states, supporting flexible and context-sensitive social inference in ecologically valid settings.

# 3 Omni-Perspective: A Scalable One-Image-For-All Benchmark From Visual Perspective to Intentionality Understanding

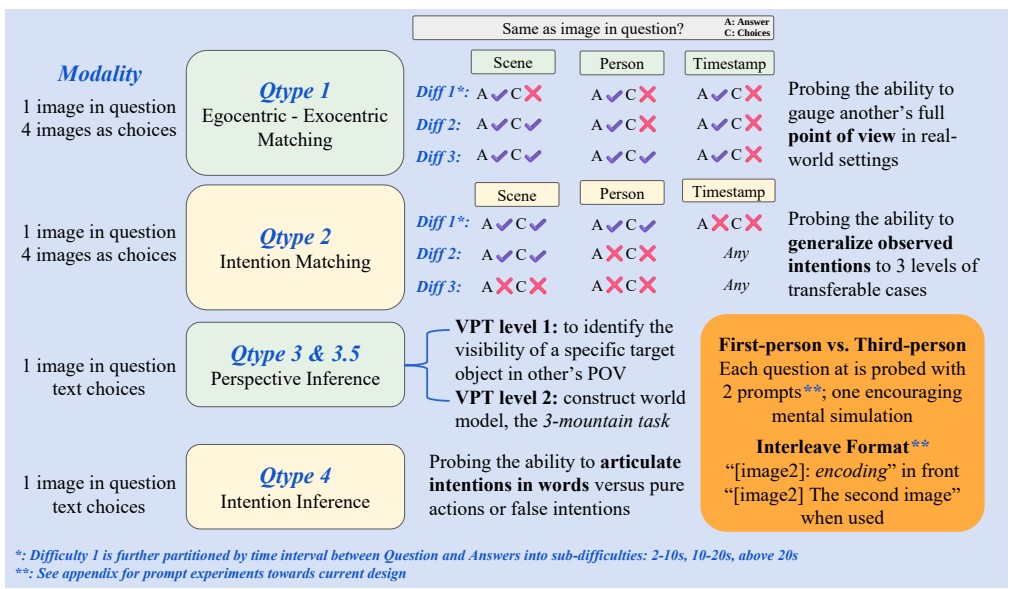

Figure 2: Overview of Omni-Perspective Bench

We define four distinct MCQ question types. Each is designed to target specific subskills aligned with the Theory-of-Mind hierarchy.

*Qtype 1 (Multi-image, Egocentric - Exocentric Matching)* - This question type presents the model with an exocentric image of a human in action and asks it to identify the corresponding view from four egocentric images. This task primarily probes Level-1 visual perspective-taking, requiring the model to reason about what the person sees based on spatial alignment and visual cues. Example prompt: "You are given an exocentric view of a person... Which of the following images best depicts what the person sees from their perspective?"

*Qtype 2 (Multi-image, Intention Similarity)* - In this task, the model is given an exocentric image of a person in action and asked to select the image depicting the most similar intention from four exocentric candidates. This question assesses the ability to generalize intention inference across individuals and scenes, contrasting with Qtype 4, which focuses on discriminating between actions

and intentions within a single context. Example prompt: "Given the image of a person performing an action... Which of the following images shows someone with a similar intention?"

*Qtype 3 & 3.5 (Single-image, Spatial Perspective Inference)* – The model is shown an exocentric image of a person and asked to determine the visibility or directional relation of an object from that person's perspective. All listed objects are visible in the scene, ensuring the task cannot be solved through simple object detection or visual salience heuristics. This task is inspired by the classic Piagetian "Three-Mountain Task" paradigm (Piaget and Inhelder, 1969), requiring the model to construct a Level-2 perspective-taking world model—that is, to represent not only what another agent sees, but how the scene is spatially organized from that agent's viewpoint. The model must perform an egocentric transformation of the scene, shifting reference frames to simulate another's first-person perspective. This demands an internal representation of spatial layout conditioned on agent pose and orientation. Example prompt: "From the perspective of the woman in the black shirt in the picture, which of the following items appears leftmost compared to the other choices?"

*Qtype 4 (Single-image, Intention Inference)* - This question presents a single exocentric image of a person in action and asks the model to choose the most likely intention from four textual options. To scale and control difficulty, distractor options are generated using a large language model (GPT-4o), conditioned on the image and atomic action annotation (See Section A.3). This format targets intention inference, requiring the model to go beyond object recognition. Example prompt: "You are given an image of a human performing an action... What do you think is their intention?"

## 3.1 Dataset Overview

**Ego-Exo4D Dataset**

We base our evaluation framework on the Ego-Exo4D dataset (Grauman et al., 2024), a large-scale, multimodal, multi-view video corpus featuring humans performing skilled activities such as cooking, bike repair, and COVID-19 self-testing. Each recording session (take) includes synchronized egocentric video from a head-mounted camera and up to four fixed exocentric views, capturing the same activity from multiple viewpoints.

The dataset is structured hierarchically across scenarios (e.g., cooking), physical settings (e.g., kitchen), takes (video sessions), cameras (time synchronized viewpoints), and annotations. Annotations include narration (atomic description of actions), procedural keysteps, and expert commentary, making it particularly suited for our use case. Our dataset includes below retrieved distribution of narrated images and goes beyond for prompt, ablation, and question evaluation analysis.

| Task Type | Total Count |
|---|---|
| Cooking | 70 |
| Covid Test | 101 |
| Bike Repair | 29 |
| **Total Tasks** | $(70 + 101 + 29) \times (3 \times 2 + 2) = 200 \times 8 = 1600$ |

Table 1: Task counts by type

**Generalization and Extensibility**

Our benchmark pipeline is designed to generalize to any dataset offering (1) multi-view video and (2) action-level annotation, e.g. the LEMMA dataset (Jia et al., 2020). This modularity enables the broader application of our framework to evaluate ToM reasoning in multimodal LLMs across diverse environments and tasks.

## 3.2 Benchmark Overview

**Scalable Ground-Truth Image-Intention Pair**

We construct a scalable set of image-intention pairs that serve as the foundation for all question types in our benchmark. Four scenarios are selected based on the number of annotated takes and coverage of non-repetitive actions. For each scenario, we define a set of high-level intentions and identify representative image frames by applying a narration-keywords-to-intentions mapping. This

mapping is then empirically refined using GPT-4o, which evaluates each image-intention pair and suggests corrections when misaligned. To minimize ambiguity, intentions that are visually similar (e.g. *install a wheel* and *remove a wheel*) or sequentially entailed (e.g. *set up test* and *perform test*) — referred to as confounding distractors — are excluded from co-occurrence within the same question. This iterative process enables scalable generation of high-quality ground-truth image-intention pairs. Refer to Section A.1 for more technical details.

**Comparability across Question Types**

*Reusing images across question types* - Each image-intention pair links to both egocentric and exocentric views that are time-synchronized within the same take. This allows the same visual context to be used for both perspective and intention questions, minimizing variability arising from differences in scene content.

*Consistent question phrasing* - We standardize the linguistic structure of prompts across all question types, avoiding shortcut through language cues. This reduces the risk of models exploiting superficial lexical patterns and promotes a fairer assessment of reasoning capabilities.

*Uniform image abstraction level* - All images are sampled from real-world video footage with similar resolution, camera specification, and background complexity. This avoids confounding effects associated with abstraction level — such as those seen when mixing synthetic, staged, or cartoon images with natural scenes — and ensures that all questions have perceptually comparable visual input.

**First- and Third-Person Language Query**

Each question type is presented in both first-person and third-person point-of-view to distinguish between two levels of perspective-taking. First-person prompts (e.g., "If you were the person in the image, what is in your line of sight?") encourage the model to take the subject's role, reflecting a mental simulation of world model and thus Theory-of-Mind reasoning (Barresi and Moore, 1996). Third-person prompts (e.g., "Given the image with a person in action, what is their intention?") treat the model as an external observer, targeting Level-1 perspective-taking.

**Distractors with Multiple Difficulty Levels or Types**

Qtype 1 and 2 in our benchmark are presented at three levels of difficulty, defined by the design of distractor choices. Difficulty increases as distractors become visually similar to the correct answer (e.g. comparable objects or spatial arrangements), while easier distractors differ more clearly in object type or environment setting. Qtype 4 does not use fixed difficulty levels but instead includes three semantically distinct distractor types, ranging from low-level action descriptions to high-level intentions. This controlled variation allows us to probe the robustness and granularity of model reasoning under varying cognitive demand.

# 4 Experiment

## 4.1 Setup

**Inference:** With the curated QCA-prompt, we assessed an extensive collection of models spanning a wide spectrum of architectures, parameter scales, and training methodologies. Our study encompassed a total of 61 MLLMs. The selection included prominent proprietary models such as those from the ChatGPT and Claude families, chosen for their established performance and widespread use. The open-source cohort featured state-of-the-art models, including InternVL, the Qwen series, and the recently released DeepSeek models, which have received increasing attention for their strong performance in multimodal tasks. The open-source models under evaluation ranged in size from 1 billion to 110 billion parameters, enabling detailed performance analysis across scales. Proprietary models were evaluated through API calls on standard personal computers. For open-source models, we performed inference locally on a compute cluster equipped with 8×NVIDIA A100 80GB GPUs. In practice, models under 13B parameters were typically executed on a single GPU, models between 13B and 32B required two GPUs, those between 32B and 70B utilized four GPUs, and models exceeding 70B ran across all eight GPUs. We adhered closely to the official inference codebases provided by model developers to ensure reproducibility and preserve model-specific inference optimizations. To further ensure consistency and correctness in handling multimodal inputs, we developed a unified evaluation toolkit capable of parsing and validating model responses across varying input formats.

**Evaluation:** To determine correctness, the model's selected option is compared against the ground truth, with any instance labeled as FAIL in the matching process automatically marked incorrect. Specifically: 1) Template aatching is attempted first, using a set of pre-defined output formats to map the model's response to one of the answer choices. 2) If template matching fails, the instance is passed to LLM matching, where a large language model—Llama-3.1-70B-Instruct(Grattafiori et al., 2024)—acts as a semantic judge to infer the intended answer choice.

To reduce the influence of answer-position bias, we adopt circular evaluation (Liu et al., 2024b). In this method, the multiple-choice options for each question are rotated across all possible positions. The model must correctly answer all k permutations of a k-choice question to be considered accurate—ensuring that its success is not due to token position or randomness.

## 4.2 Main Results

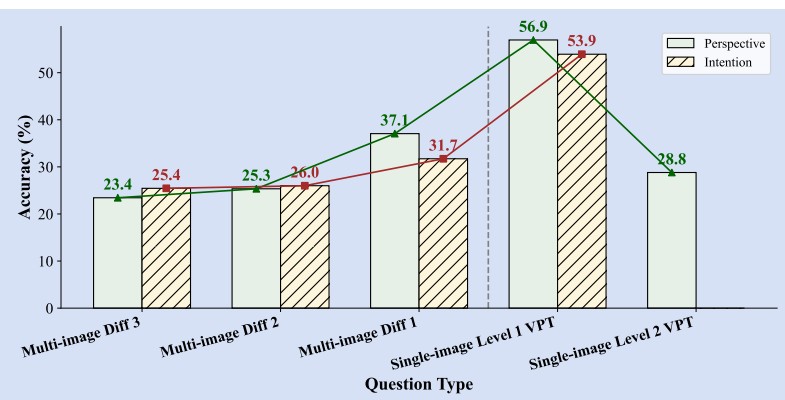

Figure 3: Comparative result between perspective taking and intention understanding across different difficulty levels and input types.

**Visual Perspective Grounding in Multi-Modal Large Language Models** We present comparative results (perspective vs. intention) across different difficulty levels (difficulty 1, 2 and 3) and input settings (single v.s. multi-image) in Figure 3. Several expected observations validate our benchmark design: 1. As difficulty increases from left to right (in the left section of the dashed line), both perspective and intention performance improve. 2. Performance on single-image tasks is consistently higher than on the three levels of multi-image tasks (to the right vs. left of the dashed line), largely due to the limited ability of MLLMs to process multi-image inputs.

Surprisingly, except for difficulty-3, where perspective is on par with intention, all other comparisons (difficulty-2, difficulty-1, and single-image) show better performance in perspective taking than in intention understanding. This contrasts with prior work Gao et al. (2025); Li et al. (2025). To further explore this distinction, we evaluate performance on level-2 perspective taking, specifically the three-mountain task (rightmost bar in Figure 3). In a fair comparison (both single-image), the three-mountain task performs lower than intention understanding, which aligns with previous findings Gao et al. (2025); Li et al. (2025). This suggests that the discrepancy between intention and level-2 perspective taking is not due to a lack of visual perspective-taking ability, but rather factors such as limited spatial reasoning in the current MLLMs.

**Does prompting for Mental Simulation help?** Encouraging mental simulation (putting oneself in another's shoes) is discussed to potentially benefit both visual perspective taking and intention understanding ability, raising an intriguing question: Does explicitly prompting MLLMs to perform mental simulation improve performance on these tasks (Barlassina and Gordon, 2017)? A drill down into single image-prompt pairs (less confounded by distractor selection methods) shows that prompting MLLMs with first-person phrasing significantly improves performance on perspective-taking tasks (p = 0.0321) on spatial reasoning, while remaining inconclusive for intention understanding.

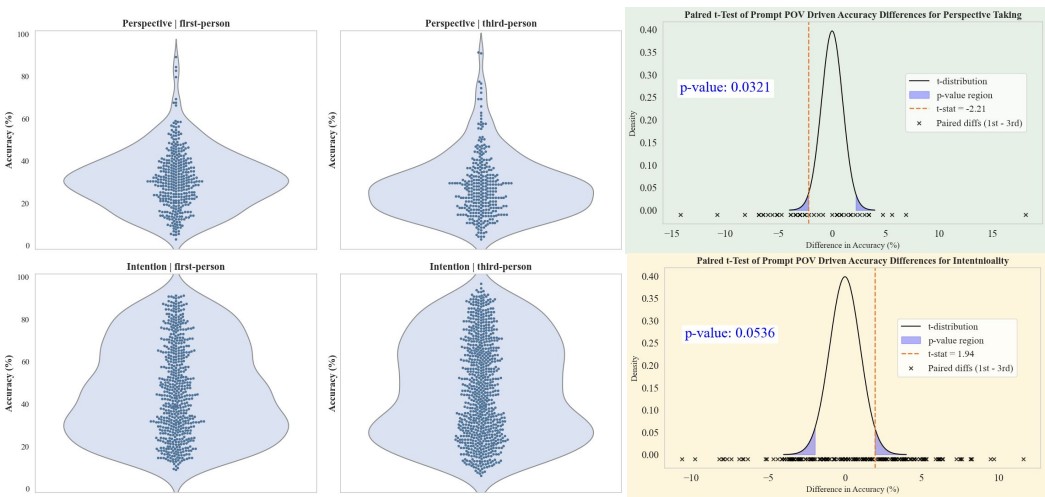

Figure 4: **Left:** Distribution of accuracy partitioned by probing concept and point-of-view of prompt; **Right:** Paired-T test results of single-image question for 2 types of prompts

### 4.3 Distractor Ablation Tests

For Qtype 4 - where distractors differ semantically (e.g. action descriptions versus high-level intentions) - we randomly select and mix choices from all three types for 200 questions. We then construct an additional ablation set of 95 randomly selected questions, each replicated into three versions containing distractors exclusively from one type. All other variables, including the image, prompt wording, and correct answer, remain constant for controlled comparison.

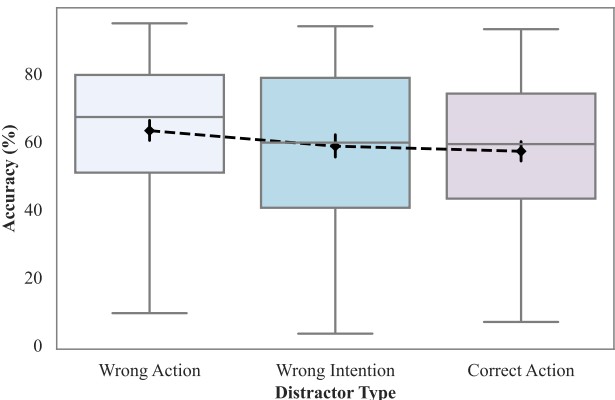

Figure 5: Accuracy by distractor type in Qtype 4 Ablation Test where the distractor type is controlled

Figure 5 reveals that average model accuracy varies across distractor types. Compared to the original Qtype 4 setup with an average accuracy of 53.9% (Figure 3), the ablation set yields consistently higher performance. This improvement likely stems from the reduced semantic variability, allowing models to exploit language-based shortcuts. Among the distractor types, wrong action results in the highest accuracy, which may be attributed to its double-layered deviation from the correct answer: it involves low-level action or object recognition rather than high-level intention inference, and the action described is itself incorrect, limiting the model's ability to rely on object-centric heuristics.

### 4.4 Benchmark Results

**Stronger Models Exhibit Greater Differentiability on Easier Tasks** Accuracy varies widely across models at lower difficulty levels, with top-performing models such as *llava-video-72b-*

| | Qtype 1 | | | Qtype 2 | | | Qtype 3 | Qtype 4 |
| | *Ego-Exo Match* | | | *Intention Match* | | | *Perspective Inference* | *Intention Inference* |
| Model | Diff1 | Diff2 | Diff3 | Diff1 | Diff2 | Diff3 | | |
|---|---|---|---|---|---|---|---|---|
| GPT-4o | 97.24% | **46.09%** | 28.09% | 75.87% | 36.28% | 30.60% | 31.37% | 59.35% |
| deepseek-vl2-small | 40.57% | 41.98% | **41.36%** | 71.81% | **73.47%** | **75.93%** | **57.08%** | 43.45% |
| Qwen2.5-VL-72B-Instruct | 95.99% | 45.05% | 35.75% | **79.26%** | 34.95% | 32.41% | 41.27% | 61.38% |
| LLaVA-Video-72B-Qwen2_multi_frame | **98.35%** | 42.69% | 29.91% | 68.62% | 35.20% | 37.96% | 46.93% | 59.23% |
| LLaVA-Video-7B-Qwen2_multi_frame | 95.28% | 38.44% | 17.99% | 67.55% | 35.46% | 41.67% | 48.11% | 51.73% |
| VILA1.5-40b | 96.46% | 32.78% | 31.78% | 56.91% | 29.34% | 23.15% | 35.38% | **75.68%** |
| Mantis-8B-Idefics2 | 75.88% | 39.25% | 28.39% | 66.57% | 37.18% | 32.33% | 32.08% | 59.85% |
| Llama-3-LongVILA-8B-256Frames | 26.18% | 29.72% | 26.87% | 59.04% | 58.67% | 58.33% | 35.14% | 73.88% |
| llava_next_interleave_7b | 67.25% | 26.55% | 21.73% | 49.71% | 27.56% | 26.72% | 34.20% | 64.38% |
| Llama-3-VILA1.5-8B | 72.17% | 28.30% | 21.96% | 40.43% | 23.72% | 23.15% | 35.38% | 60.93% |
| Ovis1.6-Gemma2-9B | 69.50% | 30.44% | 25.88% | 31.10% | 25.64% | 28.45% | 44.34% | 46.15% |
| Janus-Pro-1B | 24.76% | 26.18% | 25.23% | 43.09% | 52.55% | 56.48% | 23.82% | 32.50% |
| Vintern-3B-beta | 44.88% | 24.48% | 25.88% | 30.23% | 25.51% | 26.29% | 35.38% | 57.45% |
| InternVL2-4B | 28.38% | 24.09% | 26.63% | 37.79% | 24.36% | 23.71% | 41.75% | 51.00% |

Table 2: Accuracy by model on each Qtype subtask. Best cells are bold and both best and second-best are shaded.

*qwen2_multi_frame* achieving near-perfect scores (98% on Qtype 1 Difficulty 1), while many others remain below 30%. This variance diminishes as task difficulty increases: the standard deviation in accuracy drops from nearly 20% at Difficulty 1 to under 5% at Difficulty 3. This pattern is most evident among stronger, higher-capacity models, which show clear separation on simpler tasks but converge to similarly low accuracy as complexity rises. Weaker models, by contrast, perform consistently poorly across all levels with limited differentiation.

**Model Series Show Consistent Performance Trends** Certain model series consistently outperform others. The *qwen2_5_vl_series* and *llava_video_multiframe_series* perform especially well at larger scales, often scoring above 50% across tasks. Conversely, the *eagle_series_x4* and *x5* models underperform broadly; even the 13B variant *eagle-x4-13b-plus* averages below 20%, suggesting potential limitations in architecture, pretraining, or fine-tuning strategies.

**Scaling Model Size Yields Diminishing Returns Beyond a Point** Larger models generally outperform their smaller counterparts. For instance, in the *vila_series*, *vila1.5-40b* achieves a mean accuracy of 48%, outperforming *vila1.5-13b* (39%) and *vila1.5-3b* (33%). However, some series show marginal benefits from scaling: *llava-video-72b-qwen2_multi_frame* only slightly outperforms its 7B counterpart (52% vs. 50%), and within *internvl2_series*, the jump from 2B to 40B offers limited accuracy improvement. This suggests that beyond a certain threshold, increases in model size alone may not yield proportionate gains.

# 5 Discussion

This study introduces the Omni-Perspective benchmark, a cognitively grounded and scalable framework for probing MLLMs along the developmental hierarchy of ToM reasoning. We find that while models perform reliably on Level-1 perspective-taking tasks, they consistently struggle with Level-2 visual perspective-taking and intention inference. This pattern generally aligns with developmental theories suggesting that higher-order social reasoning builds upon more basic perceptual capacities, and is thus inherently more demanding. This suggests that MLLMs may be situated within a human-like developmental trajectory for social cognition, albeit currently limited to lower levels of the hierarchy. The observed performance gap reveals a key limitation in current MLLMs: their limited capacity for mental simulation—a mechanism believed to support flexible, context-sensitive social inference. Furthermore, our ablation studies show that model behavior is highly sensitive to distractor configurations and prompt phrasing, indicating a reliance on superficial cues rather than robust mental state representations. Taken together, the Omni-Perspective benchmark offers a controlled and interpretable framework for evaluating social reasoning in MLLMs, while also providing diagnostic insights into their architectural and training limitations.

In the meantime, we acknowledge that our benchmark relies on videos with sustained, non-transient task focus as a proxy for intentionality, which may not generalize to brief or socially nuanced intentions. It also assumes access to multiple viewpoints, limiting applicability to monocular settings.

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

# Appendices

## A  Dataset Details

### A.1  Ground-Truth Image-Intention Pair Generation

The section contains the essential information used to scale the ground-truth image-intention pair generation process. Below, we detail key design choices and procedures.

**Scenario and Task Selection** - Scenarios and tasks with repetitive behaviors (e.g., dancing, instruments playing) are excluded. The Table 3 lists all scenarios and tasks considered.

Table 3: Scenario and Applicable Tasks

| Scenario | Applicable_task_name |
|---|---|
| **Bike Repair** | Install a Wheel, Remove a Wheel, Fix a Flat Tire - Replace a Bike Tube, Clean and Lubricate the Chain |
| **CPR** | First Aid - CPR |
| **Covid Test** | Covid-19 Rapid Antigen Test |
| **Cooking** | Making Cucumber & Tomato Salad, Making Greek Salad, Making Sesame-Ginger Asian Salad, Making Chai Tea, Making a Milk Tea, Cooking Noodles, Cooking an Omelet, Cooking Scrambled Eggs, Cooking Tomato & Eggs, Cooking Dumplings, Cooking Pasta, Cooking Sushi Rolls, Cooking Samosas, Making Greek Salad, Making White Radish & Lettuce & Tomato & Cucumber Salad |

**Intention Definition and Keywords Mapping** - For each selected scenario, we define a set of high-level intentions (Table 4). We apply a two-stage matching process:

1. For each take, we extract all action-level narrations and compute cosine similarity between narration sentences and the keyword list associated with each intention (Table 5).

2. From each take, we select up to three frames (from the annotated *best_exo* camera) with the highest similarity scores for each intention, ensuring a minimum 10-second separation to avoid look-alike images. These are used as first-pass image-intention candidates.

Table 4: Scenarios and Associated Intentions

| Scenario | Intention |
|---|---|
| **Bike Repair** | Install a wheel |
| | Replace the tire tube on the wheel |
| | Clean and lubricate the chain |
| | Remove a wheel |
| **CPR** | Confirm patient consciousness |
| | Call for help |
| | Press for heart rate |
| **Covid Test** | Set up for test |
| | Understand instruction |
| | Perform test |
| **Cooking** | Prepare ingredient |
| | Preheat pan for cooking |
| | Add flavor to dish |
| | Clean up work station |

Table 5: Intention to Keywords Mapping

| Intention | Keywords |
|---|---|
| **Install a wheel** | install, attach, bike fork |
| **Replace the tire tube on the wheel** | tire level, tire valve, inflate/deflate, tire tube, bike inner tube, fit the bike tire |
| **Clean and lubricate the chain** | chain lube, degreaser spray, lubricant bottle, hold the towel, clean the chain, pick up a brush, spray water |
| **Remove a wheel** | removes the bicycle wheel, removes the wheel, take off wheel |
| **Confirm patient consciousness** | pat, check for breathing, observe, tap |
| **Call for help** | wave her hands, extend right hand, extend left hand, call for help |
| **Press for heart rate** | interlace the fingers of this hands, compress, interlock, press |
| **Set up for test** | put on desk, place on desk, pick out from box, set up, open the box |
| **Understand instruction** | test manual, test instruction, read, understand, flip |
| **Perform test** | insert test swab, pick up the collection swab, dip the swab, nostril, nose |
| **Prepare ingredient** | chopping board, tomato, onion, scallion, knife, cut, carrot, potato, banana |
| **Heat pan for cooking** | press a switch, take the skillet, turn on heat, adjust the heat, turn on gas stove, picks the frying pan |
| **Add flavor to dish** | pick up black pepper, pick up the salt, soy sauce, sauce, sugar |
| **Clean up work station** | wash, turns on the tap, opens the tap, waste bins, push dirt into sink hole, picks the dirt, trash can |

**Confounding Distractors** - As shown in Table 6, for some intentions, we define the confounding distractors that are either visually similar with or sequentially entailed to each other, and avoid presenting them within the same question.

Table 6: Intention and Confounding Distractor Pairs

| Intention | Confounding Distractor |
|---|---|
| **Install a wheel** | Remove a wheel |
| **Remove a wheel** | Install a wheel |
| **Confirm patient consciousness** | Press for heart rate |
| **Press for heart rate** | Confirm patient consciousness |
| **Set up for test** | Perform test |
| **Understand instruction** | Set up for test |
| **Perform test** | Set up for test |
| **Prepare ingredient** | Clean up work station |
| **Clean up work station** | Prepare ingredient |

**LLM Validation** - We then use GPT-4o to validate each image-intention pair.

Sample Prompt:

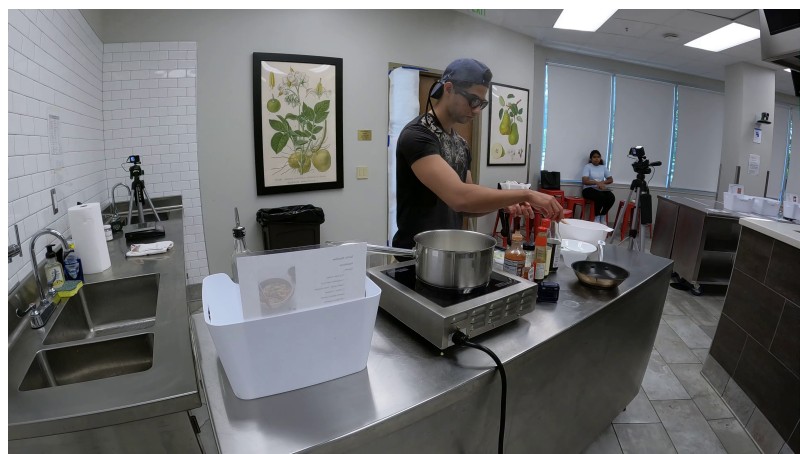

Figure 6: Sample Image Input for LLM Qtype4 Distractor Generation - Cooking

- I will provide an image of a person performing an action related to *Cooking* (*note: Scenario*), and a phrase that tries to describe the intention of the person: *"Add flavor to dish"* (*note: Intention*). Return only the required strings in a list format based on the following instructions, without additional explanations.
- Return 'great' if you are confident that the phrase accurately describes the intention of the person in the image.
- Return 'good' if you think the phrase describes the intention, but not as confidently.
- Return 'wrong' if the phrase is unrelated to the image, is not the intention that a normal non-technical human viewer could infer from the image, or has a better alternative from the following list: *[Prepare ingredients, Clean up work station, Add flavor to dish, Preheat pan for cooking]* (*note: All intentions in the scenario*).
- If you choose 'wrong', also return the best alternative option from the list. If none of the alternatives work, return 'None'.

## A.2 Qtype 3 Question Generation

We utilize GPT-3o to scale the question generation process for Qtype3. Below documents the detailed prompt we provide to the LLM.

**Context**

You will receive one or more third-person photos of everyday scenes. Each image contains:

1. a **red gaze line** that starts at the eyes of the **primary person** (the "subject"), and
2. several clearly identifiable objects.

Your task is to write **perspective-based multiple-choice questions (MCQs)** that test spatial reasoning **from the subject's viewpoint** (not the camera's).

**MCQ Templates**

- Type: Visibility - From the perspective of SUBJECT, which of the following items in the image are visible?
- Type: Direction - From the perspective of SUBJECT, in which direction is TARGET-OBJECT?
- Type: Leftmost/Rightmost - From the perspective of SUBJECT, which of the following items appears leftmost / rightmost?

Note on choices: All options must be generic and unambiguous (e.g., "a red box on the counter" rather than "a toolbox"). Label the correct answer A–D.

**Workflow**

1. **Load the image**
   - (a) Note the general setting (kitchen, bike workshop, etc.).
   - (b) Locate the subject (person with the red line).
   - (c) **Determine subject orientation** — choose exactly one:
     - facing-camera
     - back-to-camera
     - profile-left (subject looking toward **camera-left**)
     - profile-right (subject looking toward **camera-right**)

     *If the body is roughly 45°, combine them, such as facing-camera & profile-right*
   - (d) **Build a subject-centric frame**
     - **Forward** = the red gaze line.
     - **Left / Right** = rotate the frame ± 90° around the subject.

| Subject Orientation | Subject-Left | Subject-Right | Quick Visual Cue |
|---|---|---|---|
| facing-camera | camera-**right** | camera-**left** | (mirror rule) |
| back-to-camera | camera-**left** | camera-**right** | (mirror rule) |
| profile-left | **down** in photo | **up** in photo | |
| profile-right | **up** in photo | **down** in photo | |

   - **Behind** = opposite of forward.
   - If subject orientation is combined (e.g., facing-camera & profile-right), the projection should also be combined.

2. **Parse objects**
   List every salient object as *minimal-adjective + generic noun* (e.g., "blue mug," "metal faucet"). Re-use these exact names in the MCQs.

3. **Generate three MCQs (one of each type) per image**
   - Describe the subject succinctly (e.g., "the woman in a blue apron").
   - **Direction**: pick a clear {TARGET-OBJECT}; options = front / behind / left / right.
   - **Visibility & Leftmost/Rightmost**: provide four distinct objects.
   - Mark the correct answer.

4. **Quality check (mandatory)**
   - Verify every spatial relation in the subject-centric frame.
   - Ensure wording is concise, bias-free, and each referenced object is clearly visible.

5. **Output** — one JSON record per question. {
   "image_id": "<image filename or UID>",
   "subject_direction": "facing-camera | back-to-camera | profile-left | profile-right | <combined>",
   "question_type": "visibility | direction | leftmost | rightmost",
   "question": "<full question text>",
   "options": "A": "...", "B": "...", "C": "...", "D": "..." ,
   "answer_key": "A/B/C/D"
   }

## A.3 Qtype 4 Distractor Generation

The distractor generation process for Qtype 4 requires special attention due to its textual nature.

For **Wrong Intention** distractor type, we randomly sample other intentions from the same scenario, while explicitly avoiding confounding distractors (Table 6). When the number of suitable alternatives is insufficient, we supplement the set with manually created pseudo-intentions that are plausible yet not part of our dataset (e.g. *Taste the food*, *Throw away food waste*).

For **Wrong Action** and **Correct Action** distractor types, we leverage a LLM (GPT-4o) to scale generation and validation.

Sample Prompt:

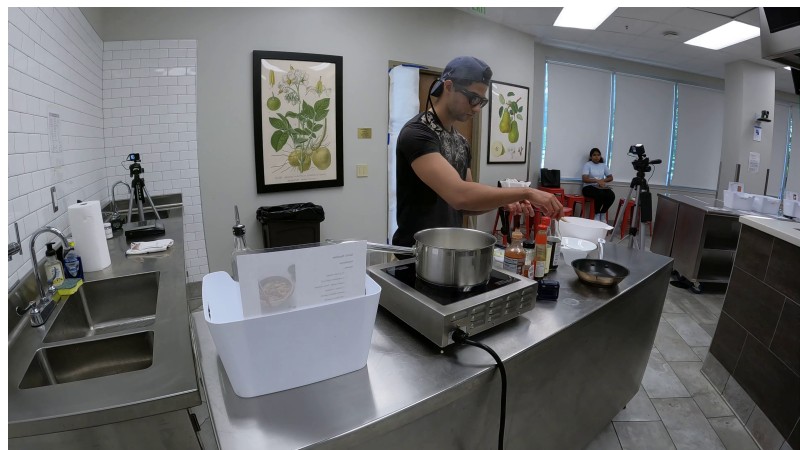

Figure 7: Sample Image Input for LLM Ground-Truth Validation - Cooking

You are an expert in linguistics and are good at coming up natural alternative expression if given a sentence in English.

Give the sentence 'C takes the dark soy sauce with his right hand.', please come up with the following, without including any explanations.

1. Type 3: 5 concise phrases that describe the action (atomic description) in the sentence. If the sentence doesn't have 'C' (a human) as the subject, make sure to phrase the action such that it sounds reasonable if the subject is a human.

2. Type 2: 5 concise phrases that describe different but similar actions. For example, these alternate phrases can EITHER a) describe the same action on a different object, OR b) describe different action on the same object. Do not replace both action and object at the same time. It is preferred that if a human is to perform these phrases, their body gestures and/or scenario will look like the original sentence.

General requests:

1. return phrases without explicit subject. For example, 'C does something' should be shortened to 'do something'.

2. the phrases should use verbs and nouns that are natural and colloquial.

3. the phrases should make sense with human as the subject, even if the subject in original sentence may not be a human. Rephrase the original sentence to human-subject first, then generate alternatives.

The output format should follow: {'type_3': [phrases1, phrases2, ...], 'type_2': [phrases1, phrases2, ...]}

Sample Output:

{'type_3': ['grab soy sauce', 'hold dark soy', 'pick up sauce', 'lift dark soy', 'take soy bottle'], 'type_2': ['grab light soy sauce', 'hold ketchup bottle', 'pick up olive oil', 'lift sesame oil', 'take vinegar bottle']}

