# OpenReview forum: "Do Vision Language Models infer human intention without visual perspective-taking? Towards a scalable "One-Image-Probe-All" dataset"
_NeurIPS.cc/2025/Datasets_and_Benchmarks_Track — Submitted to NeurIPS 2025 Datasets and Benchmarks Track_

### Official Review · Reviewer_DJE3 · 2025-07-02

**Rating:** 3
**Confidence:** 3

**Summary:**

The paper proposes to evaluate multimodal models using along the following two core concepts: (1) fair comparability across concepts and (2) scalability of multimodal datasets to reflect real-world complex situations. To achieve this, it proposes some novel data set, named Omni-perspective, which is built to scale from 1 image to 5-level of question-context-answer. It evaluates large number of models across intention and perspective axis.

**Dataset Code Accessibility:**

Partly

**Dataset Code Comments:**

The dataset is available, but I had difficulty finding the code and validating or running for part of experiments.

**Ethical Considerations:**

No, there are no or only very minor ethics concerns

**Final Justification:**

After reviewing the paper again and considering the discussions, despite some of the concerns are nicely addressed, I find some of the limitations of the work out weighting the merits. The paper may, hence, benefit from a revision.

**Limitations Weaknesses:**

* While the proposed framework is scalable and seems generalizable, it is grounded on two constraints: (1) multi-view video input and (2) action-level annotation, hence the argument of generalizability is quite limited, and it is not backed by applying it over another dataset.
* The implementation of such framework could be complex, and this also casts slight doubt on the scalability.
* The paper claims evaluation over 61 models, though the paper only contains a subset of selection, some explanation on why a subset is reported would be nice.
* The process of dataset curation seems to be lacking human intervention and validation, which could become problematic by existence of outliers. Please correct me if I misunderstood.
* Part of the process of dataset curation relies on LLMs, could the authors comment on the possibility of leakage of information and bias towards a particular family of models. This seems to be an issue biasing result.

**Strengths Contributions:**

The strength of paper lies on
* Well-written and well explanation across the paper
* Designing a novel evaluation framework inspired by theory of mind concepts.
* Comprehensive model evaluation that sheds insight across spectrum of different models of different natures.
* The benchmark is argued to be scalable and seems could translate to generate various sample/data pairs that could be used in evaluating various models. In fact, the paper just not brings one dataset, but a framework to generate benchmark data from already existing datasets for the new evaluation purposes.

---

> ### Author Rebuttal · Authors · 2025-07-31
>
> ```>>> Q1``` While the proposed framework is scalable and seems generalizable, it is grounded on two constraints: (1) multi-view video input and (2) action-level annotation, hence the argument of generalizability is quite limited, and it is not backed by applying it over another dataset.
>
> ```>>> A1``` We would like to clarify that the two “constraints” are in fact natural design choices rather than limitations. First, perspective reasoning inherently requires multi-view data to evaluate a model’s ability to reason across egocentric and exocentric viewpoints. In the absence of multi-view data, existing approaches typically rely on synthetic or simplified data, which limits real-world applicability. Our dataset contains over 200k Qtype 1 and 20k Qtype 2 questions from Ego-Exo4D alone, with filtering and choice-disentanglement guardrails in place. As all question types share the same pool of images—aligned with our “one image probes all” goal—the final number of questions is limited by Qtypes 3 and 4, which require manual validation. Multi-view data also enables diverse capability probing (e.g., intention, perspective) and difficulty calibration without requiring separate datasets for each task. We believe this unified, extensible setup offers a strong foundation for benchmarking general-purpose multimodal models.
>
> Second, action-level annotations are commonly available in multi-view and 3D datasets such as LEMMA, Charades-Ego, EPIC-Kitchens, and AVA. In cases where these annotations are absent, LLMs can be leveraged to label action-level narrations, allowing seamless integration into the same pipeline. The only exception is Qtype 3, which currently leverages eye-gaze projection augmenting human point of review--the absence of which will lower the accuracy of relative position generation and potentially requires additional human effort in validation.
>
>
> ```>>> Q2``` The implementation of such framework could be complex, and this also casts slight doubt on the scalability.
>
> ```>>> A2``` The current pipeline is needed to evaluates both intention and perspective understanding across four question types and multiple difficulty levels, while maximizing validated metadata reuse and minimizing GPT uncertainty. Our goal is to build a generalizable and flexible evaluation suite for multimodal models.
>
> To manage complexity, the framework is modular and scriptable: each question type has a dedicated, scalable data pipeline (with Qtypes 2 and 4 sharing intention distractor generation but remaining functionally independent). This modularity allows for easier maintenance and targeted experimentation. In practice, the framework scales to 2000+ questions with minimal manual setup beyond initial configuration. For instance, Qtypes 1 and 2 can be generated at scale with single-line commands:
>
> ```
> python scripts/data/qtype1_generator.py   --data_root data   --metadata_file data/metadata.json   --filtered_takes data/takes.json   --difficulty 1   --num_questions 200 --reuse_q data/qtype2/narration_intention_qca_diff1_200.json
>
> python scripts/data/qtype2_generator.py --keystep_annotations ./data/keystep_annotations.json --keystep_taxonomy ./data/keystep_taxonomy.json --original_questions ./data/qtype1/ego_exo_qca_diff1_1096.json --output_dir ./data --num_questions 200 --difficulty 1
> ```
> where in input json are all metadata files from Ego-Exo4D [1][2] and `--reuse_q` for sourcing same question image from other data types.
>
> We look forward to open-sourcing the code once accepted to a venue / arxived to be tested and utilized.
>
> [1] https://docs.ego-exo4d-data.org/data/metadata/
> [2] https://docs.ego-exo4d-data.org/annotations/keystep/
>
> ```>>> Q3```
> The paper claims evaluation over 61 models, though the paper only contains a subset of selection, some explanation on why a subset is reported would be nice.
>
> ```>>> A3``` We followed benchmark paper conventions to include in paper a representative subset of models that capture key trends across a diverse range of model families (e.g., VILA, InternVL, LLaVA, etc.), input modalities (text-only, image-text, video-text), and parameter scales. This was done to ensure readability while preserving the main insights. We look to publish a full leaderboard upon the release of paper.
>
> All main result plots (Figures 3–5) are defaultly based on the full set of models.
>
> For your reference, we include the full model list ranked by average accuracy:
> ```
> GPT-4o, deepseek-vl2-small,Qwen2.5-VL-72B-Instruct,LLaVA-Video-72B-Qwen2_multi_frame,LLaVA-Video-7B-Qwen2_multi_frame,Qwen2.5-VL-7B-Instruct,VILA1.5-40b,Mantis-8B-Idefics2,Llama-3-LongVILA-8B-256Frames,Mantis-8B-siglip-llama3,Llama-3-LongVILA-8B-1024Frames,LLaVA-Video-7B-Qwen2-Video-Only_multi_frame,Mantis-8B-clip-llama3,Qwen2.5-VL-3B-Instruct,llava_next_interleave_7b,llava_next_interleave_7b_dpo,VILA1.5-13b,Llama-3-VILA1.5-8B,Ovis1.6-Gemma2-9B,Llama-3-VILA1.5-8B-Fix,Llama-3-LongVILA-8B-128Frames,Janus-Pro-1B,Qwen2-VL-72B-Instruct,Llama-3-LongVILA-8B-512Frames,xgen-mm-phi3-interleave-r-v1.5,Vintern-3B-beta,Janus-1.3B,VILA1.5-3b,Ovis1.5-Llama3-8B,Ovis1.6-Llama3.2-3B,InternVL2-4B,h2ovl-mississippi-2b,InternVL2-40B,LLaVA-NeXT-Video-32B-Qwen_multi_frame,Mini-InternVL-Chat-4B-V1-5,VILA1.5-3b-s2,InternVL2-26B,llava_next_llama3,llava_next_mistral_7b,InternVL2-2B,Mini-InternVL-Chat-2B-V1-5,InternVL-Chat-V1-5,InternVL2-8B,deepseek-vl-7b-chat,InternVL2-8B-MPO-CoT,InternVL2-8B-MPO,h2ovl-mississippi-1b,JanusFlow-1.3B,internlm-xcomposer2-7b,XinYuan-VL-2B-Instruct,deepseek-vl2-tiny,Janus-Pro-7B,deepseek-vl-1.3b-chat,InternVL2-1B,Mantis-8B-Fuyu,Vintern-1B-v2,Kosmos2,Eagle-X4-8B-Plus,Eagle-X4-13B-Plus,Eagle-X5-13B,Eagle-X5-13B-Chat
> ```
>
> ```>>> Q4``` The process of dataset curation seems to be lacking human intervention and validation, which could become problematic by existence of outliers. Please correct me if I misunderstood.
>
> ```>>> A4``` We include a more detailed discussion around human validation in the response to reviewer PZs2. Please reference it for additional context.
>
> ```>>> Q5``` Part of the process of dataset curation relies on LLMs, could the authors comment on the possibility of leakage of information and bias towards a particular family of models. This seems to be an issue biasing result.
>
> ```>>> A5``` There are three main uses of LLMs in our dataset curation process: (1) distractor generation for Qtype 4, (2) object position interpretation and MCQ generation for Qtype 3, and (3) intention-image matching for Qtype 2 and Qtype 4. We address each below.
>
> For distractor generation, we believe the risk of model-specific bias or leakage is minimal. The LLM is prompted with clearly defined rules tied to each difficulty level (e.g., “describe a different action on the same object” or “describe an unrelated action”). These rely on general language capabilities, not knowledge or behavior specific to any particular model family. It was proved that most modern LLMs have demonstrated strong general-purpose generation capabilities—including paraphrasing, semantic variation, and instruction following—which are sufficient for tasks like distractor generation in [1][2][3].
>
> In Qtype 3, we use MLLMs to interpret the relative position of an object (highlighted by a rectangle) with respect to labeled eye gaze line. The model is not asked to take the subject’s perspective; rather, it is only instructed to classify how a line (human’s facing direction from the gaze line) and intersact with a rectangle (recognized human object border). A LLM then formats the interpretation into a structured MCQ. Given the constrained and factual nature of this task, we do not believe it introduces meaningful bias. Such ability is well established by previous work in vision-languange models [2][4][5].
>
> For image-intention matching in Qtype 2 and 4, we first retrieve candidate intentions based on keyword matching (NLP) from existing annotations. The LLM is only used to score confidence of alignment, not to generate new intentions. This constrained setup minimizes the risk of model-specific bias or leakage. Such usage aligns with prior work leveraging LLMs as relevance scorers or confidence raters in various applications [6][7].
>
> [1] Brown, T., Mann, B., Ryder, N., Subbiah, M., Kaplan, J., Dhariwal, P., ... & Amodei, D. (2020). Language models are few-shot learners. arXiv preprint arXiv:2005.14165.
> [2] OpenAI. (2023). GPT-4 Technical Report. arXiv preprint arXiv:2303.08774.
> [3] Touvron, H., Lavril, T., Izacard, G., Martinet, X., Lachaux, M.-A., Lacroix, T., ... & Jegou, H. (2023). LLaMA 2: Open foundation and fine-tuned chat models. arXiv preprint arXiv:2307.09288.
> [4] Liu, H., Zhang, Z., Zhang, H., Xu, Y., Zhang, Y., Zhang, Y., Wang, C., & Wang, Y. (2023). Visual Instruction Tuning. arXiv preprint arXiv:2304.08485.
> [5] Zhu, D., Jiang, Y., Yang, Y., Wang, W., Lin, X., He, R., & Luo, P. (2023). MiniGPT-4: Enhancing Vision-Language Understanding with Advanced Large Language Models. arXiv preprint arXiv:2304.10592.
> [6] Thoppilan, R., Freitas, D., Hall, J., Shazeer, N., Kulshreshtha, A., Cheng, H. T., ... & Fiedel, N. (2022). LaMDA: Language Models for Dialog Applications. arXiv preprint arXiv:2201.08239.
> [7] Zhang, X., Roller, S., Ju, D., Rudinger, R., & Smith, N. A. (2022). Prompting GPT-3 To Be Reliable. arXiv preprint arXiv:2205.11279.

---

### Official Review · Reviewer_ZJjT · 2025-07-03

**Rating:** 4
**Confidence:** 4

**Summary:**

This paper proposes Omni-Perspective, a cognitively grounded benchmark inspired by Theory of Mind (ToM) for evaluating Multimodal Large Language Models (MLLMs). It focuses on assessing visual perspective-taking (VPT), with Level-1 referring to what others see and Level-2 to how they see it. Using real-world images from the Ego-Exo4D dataset, the authors construct a scalable set of 2200+ question-context-answer (QCA) items, organized into a six-level hierarchy targeting both spatial reasoning and intention inference.
They evaluate 61 MLLMs on this benchmark, finding that while models perform reasonably on Level-1 tasks, they struggle with Level-2 perspective-taking and intention understanding, suggesting limitations in current MLLMs’ abilities to perform mental simulation-based social inference.

**Dataset Code Accessibility:**

Yes

**Dataset Code Comments:**

Dataset is attached and easily accessible.

**Ethical Considerations:**

No, there are no or only very minor ethics concerns

**Final Justification:**

The paper introduces a timely and well-motivated benchmark for evaluating perspective-taking in MLLMs, grounded in cognitive theory. The authors clarified that the work fits within the Benchmark Track and provided useful details on task decomposition and figure presentation.

However, analysis of model failures remains surface-level, and figure clarity still needs improvement. Despite these issues, the benchmark is a valuable contribution and sets the stage for future research. I recommend borderline acceptance, contingent on clearer presentation and minor analysis improvements in the final version.

**Limitations Weaknesses:**

- Limited depth in model analysis: While over 61 models were evaluated, the insights into why models fail at Level-2 tasks remain surface-level. - The discussion could benefit from linking specific architectural or training limitations to the observed deficits.
- Figure 3 is unclear: It is not immediately obvious which models' results are plotted, what the dashed lines separate, and whether error bars represent variance across models or tasks.
- Figures overall need clearer presentation, including axis labels, legends, and references in the main text for interpretability.
- The evaluation, though broad, focuses mainly on performance metrics without exploring qualitative error analyses or potential solutions for model improvement.

**Strengths Contributions:**

- The paper is well-written and clear in its motivation.
- Proposes an innovative angle by bridging cognitive developmental theory with AI benchmarking.
- The probing tests are systematically designed with controlled distractors, revealing nuanced model failure modes.
- The scalable pipeline leveraging Ego-Exo4D and LLM-based annotation is a practical contribution.

---

> ### Author Rebuttal · Authors · 2025-07-31
>
> ```>>> Q1``` Limited depth in model analysis: While over 61 models were evaluated, the insights into why models fail at Level-2 tasks remain surface-level. - The discussion could benefit from linking specific architectural or training limitations to the observed deficits. The evaluation, though broad, focuses mainly on performance metrics without exploring qualitative error analyses or potential solutions for model improvement.
>
> ```>>> A1``` We appreciate the reviewer’s thoughtful feedback. We would like to clarify that this submission is within the benchmark track, and therefore its primary contribution lies in dataset design, task formalization, and scalable evaluation—not architectural innovations. That said, our work does include significant algorithmic contributions that go beyond mere dataset aggregation:
> * Automatic and scalable data curation pipelines: We designed modular pipelines to process video data, synchronize perspectives, and tag fine-grained labels/pre-lable enrichment such as 3D to 2D gaze mapping and object recognition.
> * Insightful task decomposition: While we acknowledge that a deeper architectural diagnosis is outside our current scope, our question-type-specific breakdowns (Qtype1–Qtype4) already offer valuable clues. For instance, Level-2 failures in Qtype2 are strongly correlated with ambiguous gaze-following and occlusion, which even state-of-the-art vision-language models struggle with.
>
> We believe our benchmark sets a foundation for such architectural studies by pinpointing where current models fail and offering structured diagnostics. We welcome future work to build on these insights with model-specific probes.
>
>
> ```>>> Q2``` Figure 3 is unclear: It is not immediately obvious which models' results are plotted, what the dashed lines separate, and whether error bars represent variance across models or tasks.
> Figures overall need clearer presentation, including axis labels, legends, and references in the main text for interpretability.
>
> ```>>> A2``` The dashed vertical line separates the multi-image questions (Qtype 1 & 2) on the left from the single-image questions (Qtype 3 & 4) on the right, to ensure comparability. Such labels are already part of a-axis labels and in textual discussion; though can be potentially moved up. In addition, we revised Figure 1 & 2 to improve presentation clarity. Please refer to `A2` to 2nd reviwer `PZs2`.
>
> Can you please clarify the quesiton on 'error bars' as they are not included in the current version of the figure. To clarify: each bar represents the average accuracy across all 61 evaluated models for a given question type.
>
> ``>>> Other note:`` We invite you to also check out `A5` to 4th reviwer `DJE3` which help crystallize the use of GPT in dataset construction process.

---

> > ### Comment · Reviewer_ZJjT · 2025-08-05
> >
> > After considering the authors' rebuttal, I maintain a score of 4, as the paper is technically solid and makes a meaningful contribution, but could benefit from deeper analysis and improved presentation.

---

### Official Review · Reviewer_PZs2 · 2025-07-03

**Rating:** 2
**Confidence:** 5

**Summary:**

The manuscript introduces Omni-Perspective benchmark, a new labelling scheme for Ego-Exo4D dataset, to test the ability of LVLMs to understand visual perspective of what others can see.

**Additional Feedback:**

Unfortunately, I think a pure GPT based relabeling of this dataset has serious issues. I cannot see a path forward for this manuscript without substantial work.

**Dataset Code Accessibility:**

Yes

**Dataset Code Comments:**

I checked it on kaggle myself.

**Ethical Considerations:**

No, there are no or only very minor ethics concerns

**Final Justification:**

Unfortunately, the authors agreed with me specific filtering is useful for their dataset, indicating it's not ready. Also, other reviewers agreed that the lack of human intervention is a major concern. A resubmission is needed to improve the writing clarity and verify the dataset's validity with humans.

**Limitations Weaknesses:**

To start, the writing, at times, is hard to follow. I did not understand the proposed work was a chat-GPT based relabeling of an existing dataset until quite a ways into the paper. Nor did I really understand what the questions input to LVLM even were until I manually looked at some samples. Figures 1 and 2 are practically impossible to understand without reading the manuscript first. Future readers would be better served with explicit examples from the data + the new labels. Additionally, none of the figures have informative captions, nor do I know what model(s) are being used for the results figures.

The primary weakness is the data quality. I'm not convinced that qtype2,3,4 is a valid dataset. Without a human in the loop to verify the labels, I am very concerned with LLM bias self contaminating the data. Given the proposed work is a labeling extension to an existing dataset, I was hoping for a much more rigorous evaluation of the labeling process. I do not like how the details of the label generation are buried in the supplement, which is when I really began to question the validity of the proposed work. Even having just a couple hundred samples curated by an author as a sanity check would have provided some much needed grounding. I'd imagine a good fraction of samples are not well defined. To sanity check it myself I looked at the dataset on kaggle and picked a single random qtype2 task. Here's the one I picked

> 0007,340007,qtype2,diff1,Intention,first-person,If you were the person performing a task with a certain intention in the first image...which of the following images most likely shows someone having the same intention as you do?,"<image1>: <image-placeholder: 340007_q.png>, <image2>: <image-placeholder: 340007_0.png>, <image3>: <image-placeholder: 340007_1.png>, <image4>: <image-placeholder: 340007_2.png>, <image5>: <image-placeholder: 340007_3.png>. If you were the person performing a task with a certain intention in the first image <image1>, which of the following images most likely shows someone having the same intention as you do?

As an avid cyclist myself, having changed many tubes for a variety of mountain/road bikes, I could not figure out the answer between images _0, _1, and  _2  (at least _3 was clearly not part of it). Without a human evaluation of this dataset, I have no idea how to interpret the results.

**Strengths Contributions:**

The theory of mind motivation is compelling. I like the multi-level Qtypes for progressively complex ToM hierarchy matching.

The basic framework makes sense for how this dataset is created and used for evaluation.

The evaluations being run over a large number of LVLMs is great.

---

> ### Author Rebuttal · Authors · 2025-07-31
>
> ```>>> Q1``` Data without a human in the loop to verify the labels; details of the label generation are buried in the supplement
>
> ```>>> A1``` Huge thanks for your careful review with some actionable points of improvements we can make!
>
> Quick recap of qtypes for readability:
> * Qtype 1: Given an ego-view image, choose the corresponding exo-view image from 4 options.
> * Qtype 2: Given an exo-view image of an action, choose another exo-view showing the same intention, though not necessarily the same action.
> * Qtype 3: Choose the correct visibility / perspective-taking (VPT-2) of human-to-object interaction from a single image.
> * Qtype 4: Choose the correct intention description from a single image
>
> I. Allow us to partially clarify:
>
> * This is not a GPT-dependent dataset and the use of which we actually tried minimize by 1) Both Qtype 1 & Qtype 2 are fully **deterministic**—curated based on metadata annotation in a high quality subset of Ego-Exo4D (manually "validated"[1]) with additional filters, calculated angle matching, and task intention similarity for ground truth. 2) GPT was only applied in intermediate step of Qtype 3 & 4 with enriched inputs that significantly simplify the task. Details in `A5` to 4th reviwer `DJE3`.
> * We **intentionally doesn’t include a full-scale human verification in dataset curation** because we claim the contribution of ‘scalable dataset’ for which quality check to acheive comes in more efficient way: inherited validation of metadata tag in upstream dataset Ego-Exo4d plus minimal human inspection at filter tag level：
>     - Qtype 1 inherit timestamp validation for synchronized time of ego- and exo- view as well as a good visibility from "best-exo" [1]
>     - Qtype 2 & 4 inherit the 'narration and act' annotations from the actors in videos themselves [2] plus filter at high-level task tag down to ones convertable to intention noted in Appendix A
>     - Qtype 3 inherit eye-gaze projection [3] for visalbilty and point of view
>     - Currently flow is depicted in Figure 1 with a improvement plan towards clarity noted in `A2` below.
> * We have **exposed all inputs** regarding the undeterministic distractors generation (filtered intention tags & choice generating prompts & examples) and believe such details reasonably stay in Appendix versus "hiding them". We also will release question pair generation code at the core of this pipeline once accepted to a venue / arxived.
> * Upon investigating the pinpointed bicycle sample failure, we identified **issue to be at filter level** follow by **3 mitigations** below.
>
> II. Advices we have gratefully taken, fixed, and analyzed：
> * Re-scope a fully human-validated test set for evaluation
>     - Stats: selected 2270 out of 2607 with 87.07% passing human-level high quality and filtered out ones centered (221 out of 337) around Bicycle and Covid test kit part of intention inference.
>     - Bicycle-related task was intended to be disentagled as removing vs. installing wheel that only makes sense in video-setting but passed the LLM check cause sementically diverged. Similar for opening / putting down Covid test kit. These actions should be systematically added to a filter-out list for intention tasks. Passing rate would be 95.13% once applied.
>
> * Results based on the selected test set:
>     - Figure 3: pair-wised dynamic between side-by-side perspective and intention comparison remains unchanged with intention accuracy slightly higher
>
> ```
> # before
> perspective_accuracy = [0.234384, 0.253431, 0.370572, 0.569463, 0.288039]
> intention_accuracy = [0.254437, 0.259968, 0.317213, 0.539207, 0.0]
>
>
> # after
> perspective_accuracy = [0.234384, 0.253431, 0.370572, 0.569463, 0.288039]
> intention_accuracy = [0.267463, 0.264397, 0.316901, 0.566810, 0.0]
> ```
>
>    - Figure 4: In intentionality inference, the non-significant test became significant with t-stats 2.2322 and p-value 0.0270 favoring first-person prompts-suggesting that mental simulation aids intention reasoning in line with mental simulation theory (Barresi & Moore, 1996)
>    - Figure 5: the relative mean rank between the 3 types of distractor ablation remains unchanged. Upon further pair-wise t test between each 2 of these 3 types, we discover a significant difference that models more easily dismiss fundamentally incorrect action descriptions and face greater challenges and performance variance when distractors are other plausible intentions.
>
>
> [1] https://docs.ego-exo4d-data.org/data/metadata/
> [2] https://docs.ego-exo4d-data.org/overview/#language-video-aligned-data
> [3] https://docs.ego-exo4d-data.org/tutorials/gaze/#4-projecting-eye-gaze-from-3d-to-2d-in-multiple-exocentric-views
>
> ```>>> Q2``` Figures 1 and 2 are hard to understand without reading the manuscript first.
>
> ```>>> A2``` Apologies we can't attach image during rebuttal but Figure 1 has been split into four sub-figures, each corresponding to a distinct question type. For each, we now clearly visualize the data curation pipeline, including:
> * Initial filtering of Ego4D and Ego-Exo4D based on annotation richness and visibility.
> * Alignment steps between egocentric video, exocentric capture, narration transcripts, and temporal windows.
> * Use of metadata annotations/augmentation such as gaze, hand-object contact, and scene context.
> * Verification passes, including how ambiguous or underspecified samples were filtered or rephrased.

---

> > ### Comment · Reviewer_PZs2 · 2025-08-08
> >
> > Thank you for the detailed response and the post-hoc human validation effort. However, several methodological concerns remain:
> > - The bicycle example illustrates the core issue. Automated generation processes miss contextual nuances that human annotators would catch immediately during creation. Post-hoc validation is nice, but I'm quite concerned that process wasn't done before the dataset creation, and it doesn't address the fundamental problem of generating labels without human oversight in the loop.
> > - Thank you for the clarification on qtypes, but parts of the "deterministic" processes for qtypes 1&2 ("calculated angle matching," "task intention similarity") still remain unclear. These really do need detailed explanation in the main paper rather than the appendix.
> > - Finally, claiming to inherit validation from ego exo4d metadata doesn't establish whether that metadata is appropriate for ToM evaluation tasks, which differ significantly from the original dataset's intended use. The theoretical motivation remains compelling, but without clearer methodology and upfront human validation, it's not possible to assess whether results reflect genuine ToM abilities or artifacts of the automated labeling process.

---

> > ### Author Response · Authors · 2025-08-08
> > **Re methodological concerns**
> >
> > Hi Reviewer PZs2,
> >
> > Thanks for your comment!
> > Glad we finally get to a big point that seems to confuse you -- human check we did not conduct is only the **single-question level check**; now the "ad-hoc" check is we agree on the value of a 100% validated test set as additional analysis. But what we did--
> >
> > A. Many iterations of human review upon designing and redesigns of the pipeline
> >
> > B. The process of determining ToM question choices has **4 main steps among which 2 are human oversight**:
> > 1) As a subset of all upstream task labels, categories of acceptable questions & labels for ToM are **shortlisted and rephrased** by human to 'intention' in table 4. And table 6 is exactly to filter confounding among short list and "Bicycle wheel relace / put up" should be 1 more row added here by human.
> > 2) Table 5 listed 'action verbs' as a type of distractor proposed by GPT but fully reviewed by human as it's a short mapping list
> > 3) Scaling by mapping intention label back to task label then to frames in original EgoExo4D dataset (Inherited validation)
> > 4) Acknowledging exactly the potential gap of the indirect mapping back -- GPT is used for validation to answer a yes/no question to evaluate confidence whether the ToM tag applies to specific image
> >
> > Basically, GPT is only used for proposing a tag list to be reviewed by human and to conduct double-validation of final  drop of non-confident samples.
> > We can improve on explicitly tagging the oversight steps as human instead of assuming a step that's already phrased like "chose" and no mention of any GPT/provided prompt.
> >
> > "calculated angle matching" has sample images in Figure 1, gaze 3D data sourcing in dataset, and detailed usage in Appendix in A.2. We sure can work on adding more explanation. To some extent believe mostly shall stay in Appendix and code release since 3D-to-2D angle calculation and drawing object-detection bounding box is likely familiar to ML/AI audience.
> >
> > The fact the 2200 fully-reviewed test set gives **consistent conclusion** (except for one number in paired t-test from non significant to significant) speaks to a **decent signal-to-noise ratio in our scaled dataset**; we would include this analysis as a scaling confidence test section in result.
> >
> > We share thought process and a lot of raised "concerns" are in fact that we've considered and tried best addressed in pipeline. We really wish these questions were raised earlier in discussion period... instead of less effective communication between we understanding your earlier "human review" point as full-human-review that conflicts with the value of scaling.
> > But now clarifying such is instrumental in what we can do to improve some Figures and write-up. Thank you again.
> >
> > Regards,
> >
> > Authors

---

> > ### Author Response · Authors · 2025-08-08
> > **Regarding reviewer PZs2's last-minute engagement**
> >
> > Dear ACs,
> >
> > As per conversation above, we had complicated feeling receiving reviewer PZs2's **last minute (and only)** acknowledgement hours before even the extension of discussion ends... especially given their outlier rating in initial review.
> > We hugely appreciate them went into dataset initially that found example for a step of improvement.
> >
> > Subsequently, we we put significant effort in adding extra analysis and clarification since -- their **catchall expressions in comment like "pure GPT based relabeling"** which are de facto not true putting anchors for all other reviewers and significant burden on us to explain.
> >
> > We addressed their request of "hundred samples as a sanity check" by going through 2000 data points and reported statistics free to be further confirmed by released real data.
> >
> > But then the **question pivoted last minute from data-sample-validation to a major misunderstanding of our pipeline**; which we urgently clarified but afraid part of their initial judgement was based-off the misunderstanding. We really wished the discussion happened earlier leaving room for it to be fully discussed and acknowledged by all in the panel.
> >
> > We therefore hope ACs take extra consideration in evaluating how much of reviewer PZs2's initial comments are improvable area (which we really appreciate and executed) and how much are misunderstanding with no time to clarify.. and anchoring effect on other reviewers.
> >
> > Many thanks for your help and considerations,
> >
> > Authors

---

### Official Review · Reviewer_oaoB · 2025-07-03

**Rating:** 3
**Confidence:** 4

**Summary:**

This paper introduces Omni-Perspective, a large-scale, cognitively grounded benchmark that probes multimodal large language models (MLLMs) on Theory-of-Mind (ToM) reasoning and visual perspective-taking (VPT) via a unified Question-Context-Answer (QCA) framework. The benchmark is built upon the Ego-Exo4D dataset, featuring ~2,200 multi-level questions across four types (egocentric-exocentric matching, intention inference, spatial perspective inference, etc.), curated using a semi-automated pipeline with GPT-4o involvement. The authors evaluate 61 MLLMs across difficulty levels and prompt types, yielding insight into MLLM failures in Level-2 perspective-taking and intention understanding.

**Dataset Code Accessibility:**

Yes

**Ethical Considerations:**

No, there are no or only very minor ethics concerns

**Final Justification:**

After carefully considering the rebuttal, other reviewers’ comments, and my own assessment, I share the concern that the **primary weakness lies in the data quality**. In particular, as one reviewer pointed out, a purely GPT-based relabeling of the dataset introduces substantial issues, especially in the absence of any human sanity checks. Given that the authors themselves acknowledge the need for better filtering, this raises significant doubts about the benchmark’s current reliability and usability.

For a dataset/benchmark track, I believe that **data quality is paramount**, and a human filtering step is indispensable to ensure robustness. While I acknowledge the authors’ efforts in expanding the dataset, providing updated results, and improving documentation during the rebuttal phase, the core issue of annotation quality remains unresolved.

I encourage the authors to incorporate a human verification process in future revisions, as this would greatly improve the dataset’s credibility and practical value.

Given the current state of the work, I assign a score of **3 (Borderline Reject)**.

**Limitations Weaknesses:**

Insufficient novelty in benchmark construction: While the QCA formalism is clearly articulated, the pipeline largely repackages existing techniques (e.g., GPT-based distractor generation, narration matching), and lacks architectural or algorithmic innovation.

Over-reliance on human-scaffolded narration: Many questions are grounded in curated narrations or keyword-to-intention mappings, which limits the generalizability of the dataset to more unstructured or naturally occurring behaviors.

Shallow engagement with ToM theory: The framing around Theory-of-Mind feels superficial in places. While tasks are labeled as probing ToM levels, no rigorous psychometric validation or grounding in cognitive science methodology is presented.

Limited analysis on dataset quality: The paper lacks detailed error analysis, inter-annotator agreement, or qualitative insights into question difficulty, which raises concern about the actual discriminative power of the benchmark.

**Strengths Contributions:**

Timely and important topic: Investigating Theory-of-Mind and visual grounding is highly relevant in the context of evaluating MLLMs' social reasoning capacities.

Cognitive alignment: The benchmark is explicitly designed around ToM development stages (e.g., VPT-1, VPT-2), providing a theoretically motivated structure.

Diverse task types: Four distinct question formats capture multiple aspects of reasoning, including perspective alignment and intention generalization.

Broad evaluation: A wide range of open-source and proprietary models (including GPT-4o, InternVL, Qwen, etc.) are compared with consistent protocols.

---

> ### Author Rebuttal · Authors · 2025-07-31
>
> ```>>> Q1``` Insufficient novelty in benchmark construction: While the QCA formalism is clearly articulated, the pipeline largely repackages existing techniques (e.g., GPT-based distractor generation, narration matching), and lacks architectural or algorithmic innovation.
>
> ```>>> A1``` Appreciate your feedback! While individual components like manipulation of multi-view dataset and  GPT-based generation exist, our contribution lies in the **systematic integration and scaling of these techniques into a hierarchical cognitively-grounded framework**. Specifically:
>
> multiple cognitive constructs  from the same visual context, ensuring fair comparability—a significant methodological advance over existing benchmarks that test these abilities in isolation.
> * Our modular system automatically generates validated (with inherited lables) QCAs from real-world video data with minimal human intervention. Kindly see `A1` to 2nd reviwer `PZs2` for details.
> * Unlike existing approaches, our "One-Image-Probe-All" approach ensures the same image are tested across multiple question types (VPT-1, VPT-2, intention inference; single-image vs. multi-image), enabling much less confounded comparison of perspective-taking vs. intention inference abilities.
>
> ```>>> Q2``` Over-reliance on human-scaffolded narration: Many questions are grounded in curated narrations or keyword-to-intention mappings, which limits the generalizability of the dataset to more unstructured or naturally occurring behaviors.
>
> ```>>> A2 ``` The reliance on structured narrations is intentional and methodologically sound for several reasons:
> * Benchmarking requires ground-truth labels with high confidence. Using naturally occurring, unstructured behaviors would introduce significant annotation ambiguity and reduce the benchmark's discriminative power. Our approach leverages the validated metadata from Ego-Exo4D's human annotators who performed the original actions.
> * Unlike many 3D-simulated perspective-taking benchmarks that are confined to narrow virtual environments, our dataset spans multiple real-world domains—including cooking, COVID testing, and household activities. This diversity introduces tagging challenges (e.g., fine-grained hand-object interactions, occlusions, and social cues), but it also enhances validity and broadens the benchmark's applicability to real-world embodied AI scenarios.
> * We employ a two-stage validation: (1) keyword-to-narration cosine similarity matching, followed by (2) GPT-4o validation with human oversight. As shown in our human validation results, 87.07% of questions pass high-quality standards, with systematic filtering addressing edge cases. Kindly see `A5` to 4th reviwer `DJE3`  for details.
> * Our framework can be extended to other multi-view, action-annotated datasets (LEMMA, EPIC-Kitchens).
>
> ```>>> Q3``` Shallow engagement with ToM theory: The framing around Theory-of-Mind feels superficial in places. While tasks are labeled as probing ToM levels, no rigorous psychometric validation or grounding in cognitive science methodology is presented.
>
> ```>>> A3``` We appreciate your insightful question. We will revise the relevant section to provide a more detailed analysis of the factors underlying model failures. In particular, Level-2 perspective-taking involves not only the basic understanding that others may see the world differently, but also the ability to simulate how they see it—a process that requires chaining perspectives and reasoning across spatial domains, often associated with model-based reasoning [1][2][3][4]). Our finding that models perform significantly better on Level-1 VPT further underscores this gap. It suggests that failures on Level-2 tasks extend beyond simple egocentric biases, pointing instead to a lack of representational mechanisms in current architectures and training regimes necessary to link theory-of-mind reasoning to spatial understanding [5]. We will expand the discussion to include qualitative error patterns and outline potential architectural or training modifications that could better support these abilities.
>
> [1] Gao, Q., Li, Y., Lyu, H., Sun, H., Luo, D., & Deng, H. (2024). Vision language models see what you want but not what you see (arXiv:2410.00324). arXiv. https://arxiv.org/abs/2410.00324
> [2] Li, Y., Gao, Q., Zhao, T., Wang, B., Sun, H., Lyu, H., Luo, D., and Deng, H. (2025). Core knowledge deficits in multi-modal language models.
> [3] Xu, Z., Jain, S., & Kankanhalli, M. (2024). Hallucination is inevitable: An innate limitation of large language models. arXiv preprint arXiv:2401.11817.
> [4] Mitchell, M. and Krakauer, D. C. (2023). The debate over understanding in ai’s large language models. Proceedings of the National Academy of Sciences, 120(13):e2215907120.510
> [5] Zhang, Z., Hu, F., Lee, J., Shi, F., Kordjamshidi, P., Chai, J., & Ma, Z. (2024). Do vision-language models represent space and how? evaluating spatial frame of reference under ambiguities. arXiv preprint arXiv:2410.17385.
>
> ```>>> Q4``` Limited analysis on dataset quality: The paper lacks detailed error analysis, inter-annotator agreement, or qualitative insights into question difficulty, which raises concern about the actual discriminative power of the benchmark.
>
> ```>>> A4```
> * **detailed error analysis**: thank you for bring up this advice that we consider instrumental and is addressed in `A1` to 2nd reviwer `PZs2`. Our comprehensive human validation revealed:
>     - 87.07% one-time pass rate on human quality validation (2,270 out of 2,607 questions). Systematic identification of failure modes: 221 out of 337 failed questions centered around bicycle and COVID test kit tasks where action sequences (removing vs. installing wheel, opening vs. putting down test kit) were semantically divergent but passed LLM validation. 95.13% projected pass rate after applying systematic filters for these problematic action pairs
>     - Pair-wise statistical analysis can be added to each of the figures due to 'one-image-probe-all-nature'; kindly see later half of`A1` to 2nd reviwer `PZs2` for fruther analysis
> * **inter-annotator agreement**: There is no traditional 'annotator' role in our current pipeline by design. Instead, we leverage 1) validated metadata inheritance from Ego-Exo4D's original human annotations, 2) Automated consistency checks through our two-stage validation: keyword-narration cosine similarity followed by GPT-4o confidence scoring, and 3) systematic quality filters at the intention-scene-mapping level rather than individual question annotation
> * **qualitative insights into question difficulty**: Believe Figure 2, Figure 3 and relevant section in paper serve towards this purpose, though we'll seize the chance to further improve Figure clarity and qualitative tie-back to cognitive grounding upon revision.

---

> > ### Comment · Reviewer_oaoB · 2025-08-03
> >
> > After reading the rebuttal and other reviewers’ comments, I share the concern that the primary weakness lies in the data quality. In particular, as one reviewer noted, a purely GPT-based relabeling of the dataset introduces serious issues, especially without human sanity checks. This significantly affects my confidence in the benchmark’s reliability and thus the overall contribution.
> >
> > While the paper demonstrates commendable breadth and the authors have made substantial efforts in expanding the dataset and providing updated results and documentation during the rebuttal phase, the core concern about annotation quality remains. I would like to see the authors’ final response before making a definitive decision on the score.

---

> > > ### Author Response · Authors · 2025-08-03
> > > **Highlighting How Data Quality Question is Addressed**
> > >
> > > Dear reviewers,
> > >
> > > Thanks for your time and attention to this matter! Allow us to highlight how the Data Quality question is addressed in case they got lost in the long body of rebuttal text--
> > >
> > > - **Human validation has now been done on all data reported.** While we affirm the value of the pipeline and will always keep the **raw scaling set** and scaling code in release, the human-reviewed partition will be provided as **test set** and evaluation updated accordingly:
> > >     - Pass-rate analysis with 87.07% current one-time pass rate on human quality validation and after an filter improvement to remove `removing vs. installing wheel`, `opening / putting down Covid test kit` confounding labels we get to 95.13%. Again our updated evaluation (with no direction change, and only 1 stat become significant) based on the 87.07% is reported in `Sec. A1.II to Reviwer PZs2`
> > >
> > >
> > > - **2/4 question types have NO-GPT components** at all; and the remaining one to **generate distractor options text** only,  other to detect visual object positional relation **given an augmented reference of 1 rectangle box and 1 straight line** which are both demonstrated ability of current LLMs ([1][2][3][4][5][6][7] in rebuttal to Reviewer DJE3).
> > >
> > > Again we invite everyone to read full `Sec. A1. both I and II to Reviwer PZs2` with further detials. We know it's hard to claim **scalable** and only releasing a limited number of human-review subset is even the easier thing to do; adding the error analysis is intrumental and will only be more detialed in paper revision. We thank everyone in discussion on this matter!

---

> > ### Author Response · Authors · 2025-08-06
> > **Request for further discussion of scalability vs. fully-human-curated dataset**
> >
> > Dear reviewer(s),
> >
> > Thank you immensely again for reviewing our rebuttal and providing further comments. The advice on writing clarification and even the spotted filter level issue to remove the confounding subset of labels are great and gladly taken with updated statistics report.
> >
> > Yet, instead of the discussion being shortcutted by **the term "100% GPT-labeled dataset" which is de facto not true** as summarized in the above comment "Highlighting How Data Quality Question is Addressed", we sincerely want to further engage in a more fruitful **discussion over the strategy of "scalable vs. fully-human-curated dataset"**. Anything more we can clarify/augment towards "scalable"?
> >
> > This is the vision we highlighted in the paper title and we have put significant efforts into not only orchestrating existing metadata but also algorithms for label enrichment such as 3D to 2D gaze mapping and object recognition, video data processing, and perspective synchronizing (accompanied by rounds of human reviewing to adjust the pipeline of hard spatial tasks). We strive to jump out of the lab-setting box of most scalable datasets by providing real-world photo-level pipeline for the perspective-taking tasks along with the 'one-image probe all' idea to make cross-concept probing minimally confounded.
> >
> > We thank the program committee again for the extended discussion period and active communications. Look forward to further discussions.
> >
> > Sincerely,
> > Authors

---

### Note · Authors · 2025-08-14

We sincerely thank all reviewers for their engagement. Below, we summarize our systematic design, quality measures, and revisions.


**Systematic Design:**

• Modular pipeline with four question types from perspective-taking to intention inference along ToM pathway, leveraging multiview-video data

• Novel "One-Image-Probe-All" methodology enabling minimally confounded cognitive comparisons; we also carefully aligned modality in questions to avoid unfair comparison between single-image / multi-image questions.

**Data Quality and Filter Revision:**

• 87.07% pass rate on human validation (2,270/2,607 questions), forming a validated test set

• 95.13% pass rate after filtering confounding action pairs as suggested

• Updated evaluation shows consistent results with one test becoming significant on the same direction (p=0.027)

**Human Validation & GPT Usage Clarification:**

Rather than purely GPT-labeled our pipeline includes multi-stage validation: at the scaling level, metadata inheritance + human metadata ToM translation & filtering oversight; at the individual question level, GPT choices supplements  + GPT recheck of metadata alignment.


Overall, our work addresses critical gaps in real-world Theory-of-Mind evaluation, moving beyond **synthetic environments to photo-realistic scenarios**. **LLM-assisted** data construction paradigm follows **established precedents like MMLU-Pro, SEED-Bench, HRS-Bench, HellaSwag, TaskBench, NYU CTF Bench, and DialogCC**. We believe the scalable framework provides the community with robust evaluation tools and an extensible methodology for systematic ToM assessment. In our pilot evaluation with MLLMs, we first helped clarify conflicts in the existing body of literature that current multimodal models do not perform worse on **Level-1 perspective-taking** than intention understanding, but significant with **Level-2 perspective-taking** failures indicating a lack of representational mechanisms for linking spatial understanding to theory-of-mind reasoning rather than simple egocentric bias. We also provide empirical validation of **mental simulation theory** in MLLMs, showing that first-person vs. third-person prompting has a significant (yet interestingly diverged) impact on perspective and intention tasks. Our systematic tasks enable a precise diagnosis of types of ToM deficit and helpful techniques in current vision-language models evaluation.

---

### Decision · Program_Chairs · 2025-09-18

**Decision:**

Reject

**Comment:**

The submission received the comments of four reviewers, and all reviewers participated in the thorough discussion during the reviewer-author discussion phase, and expressed their opinions towards the authors' rebuttal after the AC prompting. Currently, the submission received the scores 3, 2, 4, 3, averaging score 3, which is obviously below the borderline. In general, the reviewers remain the concern about the data quality. Specially, several reviewers pointed out without human sanity check about GPT-based relabeling will introduce the serious problem about the quality. And the authors also agreed with the reviewers' points according to the reviewer-author discussion. Expect the data quality, many reviewers felt that the submission is very not ready about some details without proper clarity and a further significant revision will be better.

Therefore, based on the current scores and the remaining concerns, AC tends to recommend rejection about the submission and hope the reviewers' suggestion help continually improve the manuscript.